# BEYOND OBJECTS: CONTEXTUAL SYNTHETIC DATA GENERATION FOR FINE-GRAINED CLASSIFICATION

## ABSTRACT

Text-to-image (T2I) models are increasingly used for synthetic dataset generation, but generating synthetic training data to improve fine-grained classification performance remains challenging. Fine-tuning the T2I model with a few real examples can help generate more appropriate synthetic training data; however, this fine-tuning may also introduce overfitting and reduce diversity in the generated samples. We propose a fine-tuning strategy BOB (Beyond OBjects) for mitigating these concerns. Given a small set of real examples, we first describe them using class-agnostic attributes such as scene background and object pose. We then explicitly condition on these attributes during fine-tuning of the T2I model and marginalize them out during generation. This design mitigates overfitting, thus preserving the T2I model's generative prior and reducing estimation errors, and further minimizes unintended inter-class associations. Extensive experiments across multiple T2I models, backbones, and datasets demonstrate state-of-the-art performance in low-shot fine-grained classification when augmented with synthetic data. Concretely, BOB outperforms DataDream by 7.4% on the Aircraft dataset (from 50.0% to 57.4% when fine-tuning a CLIP classifier with five real images augmented with 100 synthetic images). Additionally, in three of the four datasets, the fine-tuning downstream models with synthetic data generated from BOB and five real images achieves better performance than fine-tuning with 10 real images. Collectively, BOB outperforms prior art in 18 of 24 experimental settings, with 2+% accuracy improvements in 14 of these settings.

## 1 INTRODUCTION

Powerful generative models trained on internet-scale datasets (Schuhmann et al., 2022; Rombach et al., 2022), more commonly called text-to-image (T2I) models, have shown promise in the creation of synthetic data for representation learning (Tian et al., 2023; 2024). However, there is still a considerable gap in performance when it comes to the creation of synthetic data which can serve as training data for downstream tasks such as classification (Burg et al., 2023; Fan et al., 2024; Geng et al., 2024), where smaller models are often used for efficiency. Ideally, given a target classification task described in language (e.g., "train a dog classifier to distinguish an Airedale terrier from a Fox terrier"), a T2I model can be directly used to generate training images of the desired classes.

One key challenge preventing T2I from generating informative images is the model estimation errors caused by a misalignment between the T2I model's learned distribution and the target task. This leads to introduction of low level artifacts and incorrect visual compositions (Geng et al., 2024). This challenge is evident in the task of fine-grained recognition. For example, the Aircraft benchmark (Maji et al., 2013) is composed of different aircraft variants. Because the images contain similar backgrounds, contextual cues are not helpful for the classification task. In addition, different aircraft types differ only on subtle details. In such cases, even minor model estimation errors can lead to performance degradation.

One approach to mitigate this is to provide a few real images per class to fine-tune the T2I model (Wang et al., 2024; Kim et al., 2024). However, operating in the few-shot regime requires special considerations (Yue et al., 2020). The increased expressivity from fine-tuning the text-encoder can introduce a trade-off where the T2I model starts to overfit to the few examples, losing its strong world prior and hurting the diversity of the synthetic dataset. These issues of overfitting are particularly

important to address in the fine-grained setting where the underlying T2I model needs to rely heavily on additional guidance from provided examples to generate not only accurate but also diverse enough examples to augment training of downstream classifier.

In this work, we tackle fine-grained classification with synthetic data generation by introducing BOB (Beyond OBjects). We leverage a captioning model to extract rich class-agnostic attributes to preserve them during fine-tuning and marginalize them out during generation. In particular, we obtain the background and pose for each example via the captioning model and add them into the text condition during fine-tuning. During data generation, we randomly sample background and pose pairs across the dataset, effectively marginalize out any unintended associations across classes.

We provide a comprehensive evaluation across three backbones, two T2I models, four datasets, two data scales, and seven existing methods, to demonstrate the effectiveness of BOB. We observe the most considerable gain on Aircraft, a dataset where T2I perform poorly and fine-tuning benefits the most. Using five real images augmented with 100 synthetic images, we fine-tune the CLIP model resulting in 7.4% increase in classification accuracy from 50.0% when augmented with DataDream to 57.4% with BOB. Further, across three of the four dataset (Aircraft, Cars, and CUB), using five real images augmented with BOB generated images results in *better* classification performance than using 10 real images: e.g., CLIP fine-tuned on CUB achieves accuracy of 75.8% with five real images augmented with BOB generated images and only 74.6% with 10 real images without augmentation. Overall, BOB outperforms existing state-of-the-art methods by at least 2% on 18 of the 24 experimental settings (backbone, dataset source, and dataset size). On the six remaining settings (on the Pets dataset), BOB offers competitive performance within 1% of state-of-the-art.

To summarize, we make the following contributions:

1. We introduce stronger supervision with more detailed captioning during T2I fine-tuning to mitigate model overfitting and enhancing prior preservation (§3.1).

2. We marginalize out unintended inter-class associations by randomly sampling class-agnostic features (background, pose) across the whole dataset (§3.2).

3. We provide a comprehensive evaluation (§4) across seven previous methods, two T2I models, and 24 different experimental settings to demonstrate our methods outperforms previous methods in 18 of the 24 settings with competitive performance in the rest.

## 2 RELATED WORK

**Personalization.** Methods for personalization aim to guide and control T2I models beyond language, typically with a few image exemplars. Many of these methods serve as the inspiration for methods specifically geared towards synthetic data generation for classification. Textual inversion (Gal et al., 2023) aims to learn a word embedding in the CLIP text encoder that can reconstruct the few images. This approach is later extended into probability space with DreamDistribution (Zhao et al., 2025). DreamBooth (Ruiz et al., 2023) fine-tunes all of the parameters of the U-Net with a rare word token. CustomDiffusion (Kumari et al., 2023) enables faster training by fine-tuning only the cross-attention layers and enables multi-concept personalization through joint training. Other recent works extends the personalization approach by learning image adapters (Ye et al., 2023; Zhang et al., 2023) or integrating a vision-language model (Li et al., 2023b; Zong et al., 2024).

**Synthetic data for classification.** Traditional data augmentation methods such as CutMix (Yun et al., 2019) and Mixup (Zhang et al., 2018) interpolates between existing data which smoothes the decision function but is limited in sample diversity and fidelity. In contrast, a pre-trained T2I model provides a world prior, and therefore, can be used to enhance both sample diversity and fidelity. Early works like Real Guidance (He et al., 2023) demonstrated that utilizing these T2I models with simple class descriptions and a few reference images can improve classification performance. Da-fusion (Trabucco et al., 2023) incorporate Textual Inversion on the few reference images to generalize to unknown concepts. Diff-Aug, Diff-Gen, and Diff-Mix (Wang et al., 2024) incorporate the additional fine-tuning of the attention in the U-Net model. Diff-II (Wang & Chen, 2025) enhances the capability during generation by interpolating between diffusion latents and augmenting prompt with image captions. DataDream (Kim et al., 2024), on the other hand, focuses on improving fine-tuning by incorporating the attention layers of the text-encoder into the fine-tuning process along with the U-Net model. Parallel to better fine-tuning, other works focus strictly on enhancing the prompt on

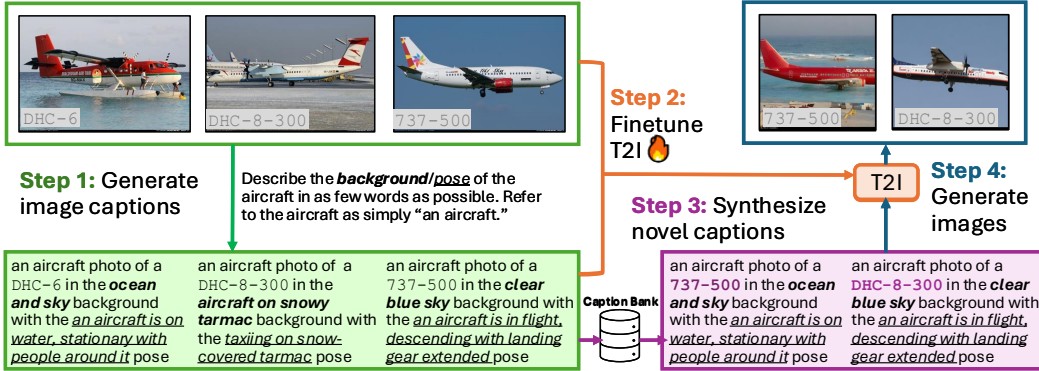

Figure 1: **Overview of BOB.** We extract background and pose attributes from training images using a captioning model (Step 1), apply **context preservation** by fine-tuning the T2I model with enriched captions containing class names and context attributes (Step 2), and then perform **context marginalization** by generating synthetic data through randomly sampling background-pose pairs across the entire dataset (Step 3-4). This preserves class-relevant features while reducing spurious class-context associations.

image side with better prompt design (Sariyildiz et al., 2023; Dunlap et al., 2023; Yu et al., 2025) or usage of image captions (Dunlap et al., 2023; da Costa et al., 2023), inverting images to the diffusion latents (Zhou et al., 2023), incorporating utilizing a vision-language model with a large language model (Michaeli & Fried, 2024) to enhance diversity of image generation, or generating hard examples (Koohpayegani et al., 2023; Hemmat et al., 2024; Askari-Hemmat et al., 2025). In contrast to previous work on better fine-tuning, our focus is on incorporating diverse captions during both the fine-tuning process and generation to generate data that is both faithful and diverse.

**Diffusion classifier.** The natural question of directly using the T2I model as a classifier emerges since training a downstream classifier is unnecessary if the T2I model itself can perform the classification task. Using the diffusion model directly for image classification has shown promising performance (Li et al., 2023a; Clark & Jaini, 2023). In addition, utilizing T2I models for classification introduces nice inductive biases such as the reduction of spurious correlations (Li et al., 2025) and better align with human vision (Jaini et al., 2024). However, the compute required to perform such classification is considerably more expensive, increasing each classification decision from seconds to over 10 minutes in the case of ImageNet. Therefore, there is still a need for research on how to distill these capabilities into a downstream classifier.

## 3 METHODOLOGY

We propose an effective approach for fine-grained synthetic data generation that addresses overfitting in T2I model fine-tuning through context preservation and marginalization, as illustrated in Figure 1. Our approach operates in both the fine-tuning stage with context preservation and the data generation stage with context marginalization. In this section, we will describe both in detail.

### 3.1 CONTEXT PRESERVATION

A key consideration in leveraging T2I models for synthetic data generation in fine-grained classification is that while these models require text-to-image mappings, classification datasets typically lack unique text descriptors for individual images. Prior approaches address this by creating class-specific text templates (e.g., "a photo of a `[classname]`"). However, such approaches reduce the rich context present in the images to a single description, and fail to preserve the diverse contextual information during finetuning. To overcome this, we propose *context preservation* that associates each image with its own unique text. Our approach extracts and explicitly encodes class-agnostic attributes (background and pose) into the text conditioning, enabling the model to learn the association between these attributes and visual context during fine-tuning. Concretely, we associate each image with a unique caption following the template: **a `[descriptor]` photo of a `[classname]` in the `[background]` background with the `[pose]` pose**.

The `[descriptor]` is a dataset-level general descriptor such as aircraft or birds. `[classname]` is the name of the class provided by the dataset. The background/pose is extracted by a captioning model for each image. We leverage the Qwen 2.5VL-7B, a state-of-the-art vision-language model (Bai et al., 2025) to extract the background with the following prompt: **describe the background of the [descriptor] in as few words as possible. Refer to the [descriptor] as simply 'a [descriptor]'**. Similarly, we use the same prompt with 'background' replaced by 'pose' to extract the pose. We also store the extracted background and pose into a caption bank $\mathcal{B} = \{(b_i, p_i)\}_{i=1}^{N}$, where $b_i$ and $p_i$ represent the background and pose attributes of the $i$-th training image, respectively (Fig. 1, steps 1-2). This prompting approach serves two purposes: (1) it provides necessary context to guide accurate attribute extraction, and (2) it prevents potential leakage of class-specific information by maintaining generic references to the object category.

Once we established the image-text pairing, we fine-tune the diffusion model using the standard diffusion objective. We follow the standard parameter-efficient fine-tuning procedure by using Low-Rank Adaptation (LoRA) (Hu et al., 2022) to fine-tune the attention layers of both the U-Net (Ronneberger et al., 2015) and CLIP text encoder (Radford et al., 2021). Consider the following notations: $\theta$ as parameters of the attention layers, the image as $x$, the text as $y$, the CLIP text encoder as $c(y)$, timestep of diffusion process as $t$, and the U-Net model as $\epsilon_\theta(x, c_\theta(y), t)$. The parameters $\theta$ are updated by minimizing the following objective: $\mathbb{E}_{(x,y)\sim D, \epsilon\sim\mathcal{N}, t\sim\mathcal{U}}\|\epsilon - \epsilon_\theta(x, c_\theta(y), t)\|_2^2$.

## 3.2 CONTEXT MARGINALIZATION

Having established a fine-tuned model that explicitly associates contextual attributes with visual context, we now leverage this learned representation to generate diverse synthetic data. The key insight is that the contextual attributes preserved during fine-tuning can be leveraged during generation to break unintended spurious class to context associations.

Consider the generation process of our training data that our T2I model emulates: image $X$ is generated given the class-relevant attributes $Y$ and the class-agnostic attributes $Z$. We introduce a random variable $I$ corresponding to a unique ID for every possible training data. The sample ID $I$ would consequentially describe the observed class-relevant attributes $Y$ and class-agnostic attributes $Z$. This generative process can be formalized as a structured causal model (Pearl, 2009) shown in Figure 2. When there are sufficient data, then $I$ becomes irrelevant because $I$ and $Y$ become independent with $P(I|Y) \approx P(I)$. Similar argument can

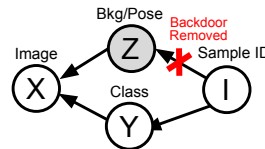

Figure 2: Causal graph of generative process.

be made between $I$ and $Z$. However, when data is scarce, the class-relevant attributes $Y$ and the class-agnostic attributes $Z$ become predictive of $I$. In consequence, this would introduce a spurious correlation: $Y \leftarrow I \rightarrow Z \rightarrow X$. These spurious correlations between class labels and contextual attributes are more prominent in fine-grained classification datasets due to the scarcity of training examples. To remove this spurious correlation and directly model the relationship between $X$ and $Y$, we would like to sample from the intervention distribution $P(X|do(Y))$ by invoking the back-door criterion (Pearl, 2009) for the following equivalence: $P(X|do(Y)) = \sum_Z P(X|Y, Z)P(Z)$.

We introduce *context marginalization* procedure to implement this principle as illustrated in step 3 and 4 of Figure 1. For each target class $c$, we generate synthetic images using the sample template structure used during fine-tuning and randomly sampling background-pose pairs $(b, p)$ from our caption bank $\mathcal{B}$ regardless of their original class associations. If we assume background and pose corresponds to the class-agnostic attributes, then randomly sampling from the caption bank $(b, p) \sim \mathcal{B}$ is equivalent to to sampling from $Z \sim P(Z)$. Afterwards, generating images conditioning on the class name as well as the sampled background and pose corresponds to $X \sim P(X|Y, Z)$. As such, our procedure approximates sampling from the interventional distribution $P(X|do(Y))$ and effectively marginalizing out spurious correlations.

## 4 EXPERIMENTS

In this section, we will go over the experimental setup for our comprehensive evaluation. Next, we will perform detailed quantitative analysis on the experimental results to demonstrate advantages of our proposed approach with respect to previous methods. Lastly, we present additional analysis

showing that our method produces synthetic data that better aligns with the target data distrbution and perform ablations showing the necessity of both the context preservation and content marginalization for generating highly informative data for the downstream classification task.

## 4.1 EXPERIMENTAL SETUP

**Datasets.** We follow the standard settings for data-scarce augmentation where we have 5 and 10 real images per class as the few-shot setting. We follow the previous evaluation setting (Wang & Chen, 2025) where we use Aircraft (Maji et al., 2013), CUB (Wah et al., 2011), Car (Krause et al., 2013), and Pets (Parkhi et al., 2012) datasets for evaluation. Additionally, we also use the CUB-LT (Samuel et al., 2021) and Flower-LT (Wang et al., 2024) dataset to extend our proposed methodology to a different data-scarce setting: long-tail classification.

**Backbones.** We use three different backbones with different degrees of language supervision during pre-training for downstream task fine-tuning. For a backbone with dense language supervision, we use the CLIP VIT-B/16 model (Dosovitskiy et al., 2020; Radford et al., 2021). For a backbone with weak language supervision, we use the ImageNet classification-trained ResNet-50 model (He et al., 2016). Finally, for a backbone with no language supervision, we use the masked auto-encoder (MAE) VIT-B/16 model pre-trained on ImageNet (He et al., 2022). *Unlike previous works which typically focus on one of these types, we decided to have all three language supervision settings to provide a wider perspective towards the behavior and usefulness of different methods.*

**Baseline methods.** We use seven popular existing data generation or augmentation methods for comparisons: RealGuidance (He et al., 2023), Da-fusion (Trabucco et al., 2023), Diff-Aug, Diff-Gen, Diff-Mix (Wang et al., 2024), DataDream (Kim et al., 2024), and Diff-II (Wang & Chen, 2025). In the synthetic data generation stage, we use the hyperparameters and procedure provided in the original paper. Since the fine-tuned T2I weights are provided by the Diff-II paper, we utilize that for synthetic data generation on Stable Diffusion v1.5 (Rombach et al., 2022). We reproduce Diff-Aug, Diff-Gen, and Diff-Mix using the same T2I model from the paper: Stable Diffusion v1.5. Similar for Datadream, we reproduce their results using the same T2I model: Stable Diffusion v2.1-base. Since RealGuidance and Da-fusion are relatively older methods on older T2I models, we reproduce their results with the more recent T2I model of Stable Diffusion v2.1-base. In summary, when possible, we utilized the relevant T2I model for each prior method to compare with directly.

**Implementation details.** For fair comparison with existing methods, we fine-tune our method on both Stable Diffusion v1.5 and Stable Diffusion v2.1-base. We utilize the same hyperparameters as DataDream with the exception of longer fine-tuning: 400 epochs instead of 200 epochs. For fair comparisons, we also extend the DataDream method to 400 epochs. We show in Section B.1 of the Appendix that both methods improve with the increased training.

In the downstream classification fine-tuning process, we replicate the few-shot examples such that there is close to a 50/50 split of real and synthetic images. The classification objective function is a weighted average on the cross-entropy loss between real and synthetic data:

$$L = \lambda \cdot CE(f_\theta(x_\text{real}), y_\text{real}) + (1 - \lambda) \cdot CE(f_\theta(x_\text{syn}), y_\text{syn}) \tag{1}$$

where $\lambda$ is a hyperparameter and $f_\theta(x)$ is the classifier with parameters $\theta$. Mixup (except for the Diff-Mix setting) and Cutmix augmentation is applied separately between the real and synthetic data. For fair comparison, we perform hyperparameter tuning by training only for 10 epochs and evaluating a separate validation set before finally training the downstream classifier with the best hyperparameters for 100 epochs. Refer to Section A of the appendix for more details.

## 4.2 FEW-SHOT CLASSIFICATION

In the few-shot classification setting we use 5 or 10 real images per class which we use to fine-tune the pre-trained T2I model before generating 100 synthetic images per class. In training the downstream classifier, the real images are replicated such that there is a 50/50 split between synthetic and real data. We present the performance of our method compared with seven existing baselines in Table 1: Diff-Aug, Diff-Gen, Diff-Mix, Diff-II using Stable Diffusion v1.5 and RealGuidance, Da-fusion and DataDream using Stable Diffusion v2.1. Downstream tasks include Aircraft classification, a task with lowest maximum starting baseline performance of 44.37%, moderate maximum baseline performance tasks of Car classification and CUB classification (79.01% and 67.72%), to

Table 1: **Few-shot classification accuracy.** The best performing method is in **bold** and the second best is underlined. Across three different backbones used as downstream classifier, our method outperforms existing methods by a considerable margin on Aircraft (AirC), Car, and CUB. On the Pets dataset, our method obtains similar performance of previous methods.

| | Method | SD Ver. | 5-shot | | | | 10-shot | | | |
|---|---|---|---|---|---|---|---|---|---|---|
| | | | AirC | Car | CUB | Pets | AirC | Car | CUB | Pets |
| **CLIP** | Real Only | | 44.37 | 79.01 | 67.72 | 92.76 | 55.73 | 84.87 | 74.59 | 93.65 |
| | Diff-Aug | v1.5 | 44.67 | 80.93 | 68.05 | 92.27 | 57.19 | 86.07 | 77.29 | 93.44 |
| | Diff-Gen | v1.5 | 47.54 | 81.60 | 69.21 | 91.69 | 58.60 | 88.43 | 76.45 | 93.57 |
| | Diff-Mix | v1.5 | 42.09 | 80.19 | 67.45 | 92.78 | 52.73 | 87.31 | 73.60 | 93.34 |
| | Diff-II | v1.5 | 49.02 | 82.16 | 70.41 | 92.75 | 60.25 | 89.02 | 77.05 | 93.02 |
| | BOB (ours) | v1.5 | 55.85 | 88.10 | **75.84** | 92.24 | **68.88** | **92.42** | **81.26** | 93.31 |
| | RealGuidance | v2.1 | 43.12 | 80.23 | 69.93 | **92.78** | 52.96 | 85.36 | 76.45 | 92.79 |
| | Da-fusion | v2.1 | 42.39 | 79.83 | 69.33 | 92.59 | 55.27 | 79.83 | 76.02 | **94.04** |
| | DataDream | v2.1 | 50.04 | 84.58 | 70.74 | 92.67 | 63.89 | 90.26 | 78.90 | 93.90 |
| | BOB (ours) | v2.1 | **57.37** | **88.41** | 75.43 | 92.73 | 67.61 | 92.00 | 80.95 | 93.77 |
| **ImageNet** | Real Only | | 39.62 | 56.16 | 48.22 | 83.17 | 55.48 | 78.50 | 68.05 | 86.75 |
| | Diff-Aug | v1.5 | 43.27 | 70.95 | 57.24 | 85.09 | 57.91 | 85.34 | 72.74 | 87.40 |
| | Diff-Gen | v1.5 | 48.42 | 80.73 | 60.91 | 86.95 | 60.32 | 88.85 | 72.40 | 89.93 |
| | Diff-Mix | v1.5 | 38.27 | 76.58 | 53.28 | 85.36 | 52.21 | 86.41 | 68.16 | 88.63 |
| | Diff-II | v1.5 | 52.28 | 82.95 | 63.60 | **87.63** | 62.81 | 88.53 | 73.60 | **89.95** |
| | BOB (ours) | v1.5 | 60.02 | **88.80** | 68.78 | 86.38 | 70.79 | 92.60 | 78.62 | 89.04 |
| | RealGuidance | v2.1 | 35.53 | 68.76 | 57.34 | 87.25 | 49.23 | 83.13 | 70.43 | 87.23 |
| | Da-fusion | v2.1 | 42.60 | 73.99 | 59.03 | 86.17 | 56.69 | 85.80 | 71.38 | 88.98 |
| | DataDream | v2.1 | 54.58 | 86.15 | 67.40 | 84.85 | 67.99 | 91.29 | 77.48 | 88.38 |
| | BOB (ours) | v2.1 | **60.31** | 88.64 | **71.38** | 87.00 | **73.78** | **92.52** | **79.52** | 89.40 |
| **MAE** | Real Only | | 41.13 | 53.94 | 39.63 | 76.81 | 57.61 | 79.12 | 62.50 | 82.97 |
| | Diff-Aug | v1.5 | 44.28 | 72.22 | 55.35 | 74.76 | 60.64 | 86.79 | 75.72 | 84.41 |
| | Diff-Gen | v1.5 | 51.79 | 82.66 | 62.79 | 77.32 | 63.85 | 90.92 | 77.10 | 85.15 |
| | Diff-Mix | v1.5 | 41.46 | 78.16 | 52.50 | 81.28 | 60.31 | 88.29 | 69.09 | 86.40 |
| | Diff-II | v1.5 | 54.90 | 82.09 | 66.53 | **88.33** | 65.20 | 90.39 | 77.05 | 89.21 |
| | BOB (ours) | v1.5 | **62.32** | 87.73 | 69.23 | 87.46 | 75.70 | **93.16** | 80.17 | **89.56** |
| | RealGuidance | v2.1 | 38.70 | 68.78 | 52.47 | 80.62 | 57.13 | 84.58 | 73.33 | 86.94 |
| | Da-fusion | v2.1 | 46.98 | 73.39 | 51.90 | 75.52 | 58.57 | 87.61 | 73.33 | 83.21 |
| | DataDream | v2.1 | 58.54 | 85.81 | 69.07 | 80.38 | 71.20 | 92.12 | 79.15 | 86.35 |
| | BOB (ours) | v2.1 | 61.21 | **88.48** | **73.21** | 86.72 | **75.85** | 92.96 | **81.29** | 88.99 |

Pets, with a relatively high maximum baseline performance of 92.76% in the 5-shot setting. Our method improves performance over all the baseline and the best performing existing method, in all tasks with the exception of Pets. For Aircraft, Car and CUB downstream tasks, BOB improves performance by at least 6.36% and up to 34.54% over the baseline of training with only the real data, and at least 2.77% and up to 10.25% over the best performing existing method. Detailed analysis focusing on specific aspects of these experiments follow.

**Aircraft classification task.** The pre-trained stable diffusion model has the least amount of knowledge about the Aircraft dataset, as indicated by the very poor performance of RealGuidance which is a personalization and fine-tuning free method. Focusing on the 5-shot setting for the FGVC-Aircraft classification task, using the ImageNet trained ResNet-50, augmenting real images with RealGuidance generated images results in a degradation in performance of the ImageNet pretrained model by 4.09% and 6.25% in 5- and 10-shot settings. Improvements by other previous methods range in 3.65-14.96% with DataDream performing the best, while our method, BOB leads to a 20.69% improvement raising the accuracy from 39.62% to 60.31%, 5.73% higher improvement than DataDream. Including the CLIP and MAE backbones for downstream tasks, BOB provides 3.78-7.33% improvement in the 5-shot and 4.65-5.79% in the 10-shot settings over the best performing previous method for this downstream task.

Table 2: **long-tail classification accuracy.** The best performing method is in **bold** and the second best is underlined. The expected accuracy across all the classes is reported. *Many* reports classes with over 20 (30) examples for CUB-LT (Flower-LT). *Medium* reports classes with between 5-20 (10-30) examples for CUB-LT (Flower-LT). *Few* reports classes with under 5 (10) examples for CUB-LT (Flower-LT). Imbalanced factor (IF) are indicated in **bold**. Results from fine-tuning an ImageNet pre-trained ResNet-50 indicates that BOB outperforms existing methods.

| Method | SD Ver. | CUB-LT | | | | | | Flower-LT | | | | | |
| | | IF=100 | | | | 50 | 10 | IF=100 | | | | 50 | 10 |
| | | Many | Med | Few | All | | | Many | Med | Few | All | | |
| Real Only | | 86.00 | 65.22 | 17.84 | 37.73 | 49.32 | 60.09 | 99.45 | 97.70 | 60.74 | 72.08 | 87.41 | 93.70 |
| Diff-Gen | v1.5 | 87.22 | 68.69 | 26.06 | 43.95 | 59.47 | 67.78 | 99.79 | 96.71 | 71.25 | 79.17 | 92.93 | 95.12 |
| Diff-Mix | v1.5 | 87.70 | 73.12 | 32.76 | 49.46 | 60.61 | 67.06 | 99.61 | 98.47 | 73.17 | 80.93 | 91.99 | 94.77 |
| Diff-II | v1.5 | 87.54 | 72.16 | 44.05 | 56.10 | 64.52 | 70.28 | 99.82 | 98.45 | 79.51 | 85.35 | 95.20 | 97.62 |
| BOB (ours) | v1.5 | **88.48** | 75.37 | 52.24 | 62.19 | 70.57 | 74.54 | **100.0** | 98.56 | 84.13 | **88.60** | 95.68 | 96.13 |
| DataDream | v2.1 | 87.25 | 71.23 | 39.72 | 53.42 | 66.05 | 72.32 | 100.0 | **98.67** | 79.96 | 85.73 | 94.13 | 96.08 |
| BOB (ours) | v2.1 | 88.43 | **75.56** | **53.47** | **63.06** | **73.00** | **76.28** | 99.45 | 98.41 | 83.48 | 88.07 | **96.85** | **97.80** |

**Pets classification task.** Pets classification task has the highest baseline performance of 76.81%, 83.17% and 92.76% with five real images and 82.97%, 86.75% and 93.65% with 10 real images for the MAE, ImageNet and CLIP backbones indicating that this downstream dataset distribution is represented much better in these backbone models compared to other datasets. We note two interesting observations. First, it appears that the Stable Diffusion model have additional knowledge of this dataset since RealGuidance can improve performance by more than 4%. Second, there is low variability in performance across all the methods in CLIP and ImageNet backbones. This is most likely due to significant overlap from the pre-training data. We comment on this in more detail in Section B.3 of the Appendix. Overall, our method BOB improves performance on par with existing methods in this task reaching a performance of 87.46%, 87 and 92.73% in the 5-shot and 89.56, 89.40 and 93.77% in the 10-shot setting, within 1% of performance of best existing method.

**Comparison of 5-shot and 10-shot performance.** An interesting comparison is to look at the informativeness of synthetic data with respect to additional real images. We observe that, the addition of synthetic generated by funetuning with 5 real images to these images using BOB outperforms using 10 real images in all datasets with the exception of Pets, indicating that BOB allows for efficient sampling of training data for finetuning the target downstream task.

## 4.3 LONG-TAIL CLASSIFICATION

To demonstrate our method extends beyond the few-shot classification setting, we perform experiments in a long-tail classification setting using the CUB-LT dataset (Samuel et al., 2021) and Flower-LT (Wang et al., 2024) using a ImageNet pre-trained ResNet-50 backbone. In the long-tail classification settings, the number of images per class used for fine-tuning is artifically skewed to follow an exponential distribution specified in Samuel et al. (2021) for CUB-LT and Wang et al. (2024) for Flower-LT. For synthetic data generation, we set a budget of 200 total images per class. Similar to the few-shot examples, we duplicate the real examples by a constant factor $c$ such that the number of images in the head (classes with abundant real images) are close to 200 images, arriving at the number of images defined by the following equation: $200 -$ number of real images $\times c$ ($c$ is 6 for CUB-LT and 5 for Flower-LT). Table 2 summarizes results where BOB outperforms existing methods by a considerable margin in the long-tail classes.

**Performance gains on CUB-LT.** CUB-LT is a relatively challenging datasets for long-tail classification with relatively lower accuracies when training with only real data. In this challenging setting, we observed across every imbalanced factor on CUB-LT datasets, our method BOB outperforms existings by a margin of at least 4%. We observe the greatest improvement in performance when there is a large class imbalanced. For the setting with the largest imbalanced (IF=100), our method improves from previous method by at least 6%: 56.10% with Diff-II to 62.19% with ours and 53.42% with DataDream to 63.06% with ours. The source of the improvement is from the im-

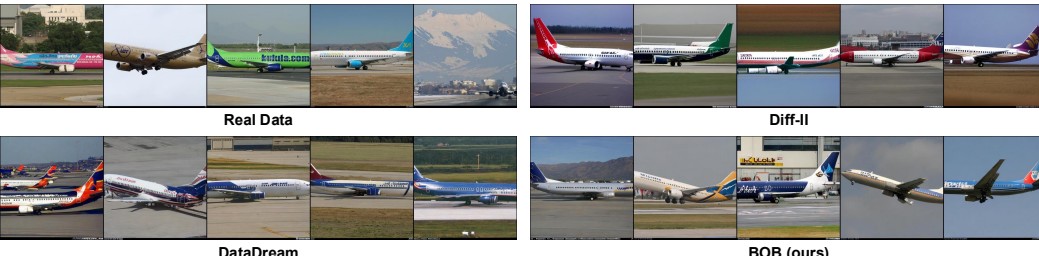

Figure 3: **Visualizations.** *left.* 737-400 images from real data and synthetic data generated by Diff-II, DataDream, and BOB (ours). Diff-II generates images with aircrafts with high contrast in simple backgrounds. DataDream generates more realistic aircrafts that are only on the ground. Our method BOB generate realistic aircrafts in very diverse settings such as taking off, flying, or on the ground with mountainous background, resulting in images that are visually similar to real images.

proved accuracy from classes with very few real examples. While classes with many examples tends to have similar accuracies (87.54% with Diff-II vs. 88.48% with ours), there is a large increase in performance in the classes with few examples: 44.05% with Diff-II to 52.24% with ours. These findings provide strong evidence that for a difficult long-tail task such as CUB-LT, BOB outperforms previous methods for synthesizing informative data.

**Competitive performance on Flower-LT.** In contrast to the dataset CUB-LT, Flower-LT is a relatively easier task with fewer classes and higher accuracies. For the most difficult part of this benchmark with IF=100, our method achieves a 2%-3% gain. For IF=50 and IF=10, our method performs competitively against previous method with accuracies within 1% range. These results show that even for easier long-tail tasks, BOB generates the best synthetic datasets.

### 4.4 ANALYSIS

Having demonstrated that our proposed method outperforms existing methods for few-shot classification and long-tail classification, we would like to understand why our method performs better. We perform analysis using the 10-shot Aircraft data setting with ImageNet-1K pre-trained ResNet-50 backbone to reveal the following: (1) our method produces synthetic data that more closely resemble and align with real data. (2) the performance gains is not due to distilling class knowledge from the captioning model, and (3) both context preservation and marginalization are important for creating high performing data.

**Qualitative analysis.** Visualization of images generated by our method shown in Figure 3 produces a sharp contrast compared to existing methods. We observe in Figure 3, previous methods either lacks realism or diversity. With Diff-II, the aircrafts have high contrast with background that are typically monotonous. DataDream generates realistic looking images with many complexities in the background but the aircrafts are all on the ground. In contrast, our method BOB, produces images that are both realistic and diverse: we have that is on the ground with a mountain in the background, an aircraft taking off, etc. If we compare with images from the real dataset, it is clear the synthetic images generated from BOB resembles the closest: suggesting that perhaps the source of performance gain is better alignment with the real data distribution.

**Real vs. synthetic distribution.** We provide an analysis of how well the synthetic dataset distributions align with target dataset by computing the per-class Frechet Inception Distance (FID) (Heusel et al., 2017) between the whole training dataset and the synthetic datasets generated by either of Diff-II, DataDream or BOB. A lower value indicates that the generated dataset is closer to the real data distribution. We reveal that the synthetic dataset produced by our perform are better aligned to the training data distribution compared to DataDream and Diff-II. In Figure 4, we observe that, on average, the per-class FID is lower for our method BOB with the mode at around 26 compared to 31 with DataDream and 37 with Diff-II. In fact, of the 100 classes in aircraft, 91 of them exhibited a

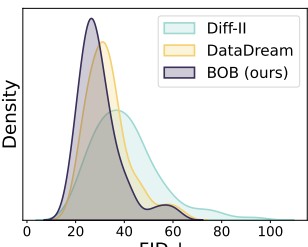

Figure 4: Density plot of FID of synthetic data against the real data for each class.

decrease in FID using our method BOB compared to DataDream (see Section B.2 of the Appendix for more details). The lower FID further reinforce our qualitative analysis that the generated data is closer to the real data distribution.

**Is it distillation?** Since the captioning models themselves have some fine-grained classification capabilities, the nature question on whether we are inadvertly distilling these capabilities down to the T2I model, and subsequently, to the downstream classification model. To test this hypothesis, we use two additional captioning model: Qwen VL2.5-3B (Bai et al., 2025) and GPT-4o (Hurst et al., 2024). If the source of performance gains is due to distillation, then the fine-grained classifcation capabilities of these captioning model should be a strong indicator of downstream classification performance. However, Figure 5 demonstrates that there is no such association at any discernable level. We observe that GPT-4o has considerably greater fine-grained classification capability with accuracy close to 80%. Yet when used as a captioning model in our method, the downstream classification performance does not improve. Similarly for the weaker captioning model Qwen-3B, we do not exhibit a considerable decrease with accuracy 70.88%, which is still over 3% higher than the DataDream baseline. All in all, this analysis suggests that the performance gains we observed is not due to distilling classification capabilities from the captioning model.

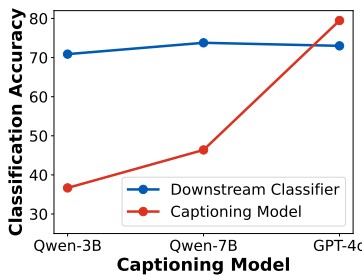

Figure 5: Classification accuracy of caption model vs. downstream classifier trained on synthetic data from BOB.

Table 3: Ablation on the effect of context preservation and context marginalization.

| marginalization | w/o preserv | w/ preserv |
|---|---|---|
| without | 68.00 | 65.90 |
| class-level | 68.01 | 64.38 |
| dataset-level | 70.13 | 73.78 |

**Ablation studies.** Finally, we would like to ablate on context preservation and context marginalization by directly including and excluding them from the pipeline. Results from Table 3 reveals three key findings. First, marginalization is necessary for generating highly informative images. Without preservation and marginalization, the algorithm is identical to DataDream baseline, which achieves only 68%. Adding marginalization without preservation result in a 3% improvement in accuracy to 70.13 %. Including both marginalization and preservation results in the best accuracy at 73.78%. Second, preservation without marginalization might lead to worse performance. We observe that the performance decreased from 68% to 65.90% when using preservation without marginalization. Finally, we highlight the need for dataset-level marginalization. To accomplish this, we add an option of performing class-wise marginalization, where we only sample background and pose from images of the same class instead of across the entire dataset. We observe that without preservation, this had no impact on the downstream performance. With preservation, this results in a *decrease* in performance from 65.90% to 64.39%. The decrease in performance is likely due to further exacerbation of spurious correlation in few-shot setting. In summary, these ablations reveal the necessity of both context preservation step and dataset-wide marginalization step for generating highly informative images.

## 5 CONCLUSION

We introduce BOB as a fine-tuning strategy for text-to-image (T2I) models that mitigates overfitting and preserves the strong world prior of these models while addressing the unique challenges of fine-grained classification. By leveraging more detailed captioning to extract class-agnostic background and pose information, conditioning on these features during fine-tuning, and marginalizing them out during data generation, our approach reduces unintended class associations and narrows the distribution gap between synthetic and real data. Extensive experiments across multiple backbones, datasets, and scales demonstrate consistent and significant performance gains, including over 7% improvement on the Aircraft dataset and state-of-the-art performance in nearly all settings. This work highlights the potential of caption-guided fine-tuning to improve synthetic data quality for downstream classification tasks and opens avenues for further research on scaling this approach to broader domains and modalities.

## REPRODUCIBILITY STATEMENT

We have taken extensive steps to ensure the full reproducibility of our work. Our methodology is thoroughly detailed across the paper, with a complete breakdown of our diffusion model finetuning, prompt generation, image generation and downstream model finetuning pipeline in Section 3.

In Section 4.1 of the main paper and Section A of the Appendix, we specify our training setups, model configurations, and evaluation protocols, including hyperparameters and dataset specifics. We utilize well-known, publicly available backbone models: CLIP ViT-B/16 (Dosovitskiy et al., 2020; Radford et al., 2021), ImageNet-trained ResNet-50 (He et al., 2016), MAE ViT-B/16 (He et al., 2022), and baseline diffusion models Stable Diffusion v1.5 and v2.1 (Rombach et al., 2022). In addition, we describe in detail our implementations of the baseline methods where we took extensive steps to ensure fair comparison. We aim to enable researchers to not only reproduce our findings but also build upon them for future research. We plan to release our complete codebase at the time of camera-ready preparation of our paper. This includes code for the entire pipeline of our method from prompt generation, T2I model finetuning, to image generation, and downstream model training.

## ETHICS STATEMENT

We use publicly available datasets Aircraft (Maji et al., 2013), CUB (Wah et al., 2011), Car (Krause et al., 2013), Pets (Parkhi et al., 2012) as well as CUB-LT (Samuel et al., 2021) and Flower-LT (Wang et al., 2024) datasets for non-commercial research purposes, in line with dataset disclaimers (e.g. CUB). These datasets may include biases in representation as the data is limited in terms of their geographical or environmental context, and the models we train with this data are not intended to be used directly without such representation considerations.

The goal of this research is to develop a method for augmenting training for fine-grained classification especially in the few-shot regime. Such settings may naturally include biases in the data due to low sample size and the downstream effects of such use must be studied in detail when applying to real-world applications. In addition, since any such method can be misused to violate privacy and copyrights or creating deepfakes, we encourage use of watermarks for both real images as well as model generated images for their detection and mitigation.

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

# Appendices

Table 4: **Hyperparameters for fine-tuning.** Hyperparameters used for fine-tuning of T2I model and downstream classifier. A list of parameters indicates the hyperparameter sweep using the validation set. The number of epochs indicated in *parenthesis* is the epochs used for fine-tuning on test set with the best hyperparameter on validation set.

|  | T2I | CLIP | ImageNet | MAE |
|---|---|---|---|---|
| Learning rate | 1e-4 | [1e-4, 1e-5, 1e-6, 1e-7] | [1e-3, 1e-4, 1e-5] | [1e-3, 5e-4] |
| Weight decay | 1e-2 | [5e-4, 1e-4] | [0.01, 1e-4, 0] | 0.05 |
| Layer decay | - | - | - | [0.65, 0.75] |
| $\lambda$ | - | [0.5, 0.8] | [0.5, 0.8] | [0.5, 0.8] |
| Epochs | 400 | 10 *(100)* | 10 *(100)* | 10 *(100)* |
| Batch size | 80 | 64 | 64 | 64 |
| Scheduler | Cosine | Cosine | Cosine | Cosine |
| Warm up | 100 steps | 3 epochs | 3 epochs | 5 epochs |
| Max norm | 1.0 | - | - | - |
| LoRA rank | 16 | 16 | - | - |
| Mixed precision | No | fp16 | fp16 | fp16 |

# A    ADDITIONAL IMPLEMENTATION DETAILS

## A.1    HYPERPARAMETER SWEEP

In this section, we go over the hyperparameter used to produce the results in Table 1 and Table 2. The full hyperparameters are listed in Table 4. For fine-tuning the T2I model on the few-shot or long-tail images, we follow the procedure in DataDream paper (Kim et al., 2024) with two difference: using dense captions for text input following our template outline in Section 3.1, increasing the number of epochs from 200 to 400. For CLIP fine-tuning, we follow the pipeline in DataDream paper (Kim et al., 2024) We optimize LoRA layers in both the image and text encoder with rank 16. We sweep over learning rate {1e-4, 1e-5, 1e-6, 1e-7} and weight decay {5e-4, 1e-7}. The only difference is that we have an additional sweep for $\lambda$ {0.5, 0.8}. This results in 16 different configurations we are sweeping over for CLIP fine-tuning. For ImageNet fine-tuning, we sweep over learning rate {1e-3, 1e-4, 1e-5}, weight decay {0, 0.01, 1e-4}, and $\lambda$ {0.5, 0.8}. This results in 18 different configurations for ImageNet. For MAE fine-tuning, we use the fine-tuning recipe provided by the original authors. We sweep over base learning rate {1e-3, 5e-4} and layer-wise learning rate decay {0.65, 0.75} and due to a discrepancy in the values provided by the original paper and default in their Github release. We also sweep over $\lambda$ {0.5, 0.8}. In all, this results in 8 different configurations for MAE fine-tuning.

We fine-tune the model for 10 epochs and select the configuration that results in the best validation accuracy across the 10 epochs. Using the hyperparameters that gave the best accuracy on the validation set, we fine-tune the pre-trained model again from scratch for 100 epochs. We report the test accuracy for the epoch that corresponds to the best validation accuracy during this training.

## A.2    PARAMETERS FOR DATASET GENERATION

We follow the same parameters used in DataDream for generating the synthetic dataset shown in Table 5: guidance scale of 2.0, 50 inference steps, and fp16 mixed precision. The scheduler used is the default for Stable Diffusion v1.5 and Stable Diffusion v2.1-base.

**Other methods.** For generating synthetic data for baselines used for comparisons, we use the default parameters used for data generation provided by their paper. For the Diff-Mix method, an additional CLIP filtering is used to remove problematic images as outlined in the original paper.

Table 5: Generation parameters.

| Hyperparameter | Value |
|---|---|
| Guidance scale | 2.0 |
| Number of steps | 50 |
| Mixed precision | fp16 |

### A.3 CREATION OF VALIDATION DATASET

For fair comparison, we perform hyperparameter tuning on the learning rate, weight decay, and the $\lambda$ hyperparameters when evaluating downstream classification performance. In order to achieve this, we created our own validation sets. In the few-shot classification setting, since not all of the training data is used, we randomly select 16 non-overlapping images per class for FGVC-Aircraft and Oxford-Pets, 10 random non-overlapping images for Stanford-Cars and CUB. However, in the long-tail setting (CUB-LT and Flower-LT), there exist some classes where most of the examples are used for training. Therefore, it is no longer possible to create a separate validation set. Therefore, we split the test set into a smaller test set and a held-out validation set with five images per class.

## B ADDITIONAL ANALYSIS

### B.1 NUMBER OF TRAINING EPOCHS

In comparison to DataDream, we increase the number of epochs from 200 to 400. To motivate this design decision, we fine-tune a model for 2,000 epochs on 10-shot Aircraft and save the intermediate checkpoints to study the effect of longer fine-tuning towards the generation of informative samples. For the intermediate checks, we follow the same synthetic data generation and hyperparameter tuning procedure to obtain the final test accuracy. The result shown in Table 6 shows similar

Table 6: Fine-tuning the T2I model longer helps.

| Epochs | DataDream | BOB (ours) |
|--------|-----------|------------|
| 200    | 66.49     | 68.65      |
| 400    | 68.87     | 74.41      |
| 1000   | 69.11     | 75.22      |
| 2000   | 67.28     | 73.87      |

effect from number of epochs on both our method BOB and DataDream. Going from 200 epochs to 400 epochs, DataDream performance improves by 2%, from 66.49% to 68.87%. However, our method exhibits a *6% increase* from 68.65% to 74.41%. The considerably larger increases suggests that, while DataDream benefits from longer fine-tuning, our method BOB benefits from it more. Similarly for both methods, the performance peaks at the checkpoint fine-tuned for 1,000 epochs before it starts to decrease again using the 2,000 epoch checkpoint. Finally, at every epoch in Table 6, our method BOB outperforms DataDream.

### B.2 CLASS DIFFERENCES IN REAL VS. SYNTHETIC DISTRIBUTION

Following our analysis in Section 4.4 towards comparing the distribution of sythetic dataset against the real dataset, we directly compare the FID of each class between synthetic data generated by DataDream and our method BOB. We plot the histogram in Figure 6. We observe that there are for 91 classes out of 100 FID is lower for our method compared to DataDream. Of these, for 29 classes, FID decreases by over 5. For all of the classes where our method had a higher FID, the increase is less than 5. This means that there are 3x as many classes that observed a considerable decrease in FID than the classes with a relatively low FID increase. This result suggests that our method provides a fairly uniform improvement in FID across all the classes-FID either a considerable increase or remain similar (within 5 FID).

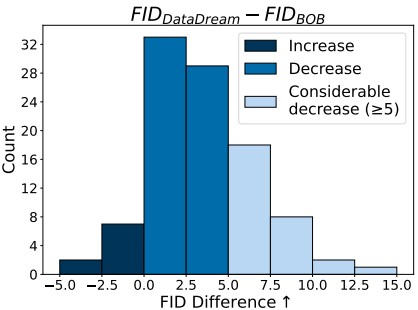

Figure 6: Histogram of the FID difference from DataDream vs. BOB for each class.

### B.3 SIGNIFICANT OVERLAPS IN THE PETS DATASET

Recall from Table 1 that on the Pets dataset, fine-tuning a classifier on synthetic data generated images results in very little performance gains across all of the methods in the CLIP and ImageNet classification pre-trained ResNet-50 backbone. In the case of CLIP, there is *no performance gains* with all of the methods arriving at an accuracy within 1% compared to just fine-tuning on real data. This is because the zero-shot classification accuracy on Pets using CLIP is already 91% as reported in the DataDream paper (Kim et al., 2024). As a result, the model already have very strong classification capabilities, and therefore, additional synthetic data isn't as impactful, if at all. For

the ImageNet classification trained backbone, we make a similar observation where most of the 39 pet classes are already present in ImageNet. To study this, we manually search up the pet names (as well as adjacent names since same pets have multiple names) in the ImageNet classes. The result is shown in Table 7. We observe that 22 of the 39 classes have a corresponding ImageNet class. Similar to the CLIP setting, if the backbone very high classification capabilities, then it is not a good evaluation metric for determining the strength of classification signals in the synthetic dataset. These findings explains why there the different trends observed in the MAE setting vs. ImageNet or CLIP setting from the Pets dataset in Table 1.

Table 7: Oxford-IIIT Pets classes with ImageNet IDs (– if not present)

| Pet name | ImageNet ID | Pet name | ImageNet ID |
|---|---|---|---|
| Abyssinian | – | Bengal | – |
| Bombay | – | Birman | – |
| British Shorthair | – | Maine Coon | – |
| Persian | n02123394 | Egyptian Mau | n02124075 |
| Ragdoll | – | Russian Blue | – |
| Siamese | n02123597 | Sphynx | – |
| Boxer | n02108089 | Keeshond | n02112350 |
| Havanese | – | Basset Hound | n02088238 |
| English Setter | n02100735 | Miniature Pinscher | n02107312 |
| Chihuahua | n02085620 | Great Pyrenees | n02111500 |
| German Shorthaired | n02100236 | Beagle | n02088364 |
| Staffordshire Bull Terrier | n02093256 | English Cocker Spaniel | n02102318 |
| New Found Land | n02111277 | Pomeranian | n02112018 |
| Leonberger | – | American Pit Bull Terrier | – |
| Wheaten Terrier | n02098105 | Japanese Chin | n02085782 |
| Samoyed | n02111889 | Scottish Terrier | n02097298 |
| Shiba Inu | - | Pug | n02110958 |
| Saint Bernard | n02109525 | | |

## C    USAGE OF LARGE LANGUAGE MODELS (LLMS)

During the preparation of this work, the authors utilized Large Language Models (LLMs) to support various aspects of the research and writing process. The specific applications of LLMs in this paper are outlined below:

**Code Generation**: LLMs were employed to generate code for visualization shown in Figure 4,5, and 6.

**Manuscript Preparation**: LLMs were used to format the LaTeX tables and figures and to help create structured appendix templates.

**Writing Assistance**: LLMs were queried to enhance clarity and readability of the manuscript. It was mostly used to generate better word choices and minor grammar improvements. All technical claims and interpretations were authored and verified by the researchers.

