# OpenReview forum: "Beyond Objects: Contextual Synthetic Data Generation for Fine-Grained Classification"
_ICLR.cc/2026/Conference — ICLR 2026 Conference Withdrawn Submission_

### Official Review · Reviewer_DX7h · 2025-10-22

**Soundness:** 3
**Presentation:** 3
**Contribution:** 2
**Rating:** 6
**Confidence:** 5

**Summary:**

Authors focus on generating effective synthetic training data, given few-shot examples of the target classes. They build upon existing methods which fine-tune generative models with the few-shot data, to also consider and marginalize out class-agnostic attributes. Specifically, they focus on background and pose. To do this, they identify sets of backgrounds and poses with Qwen2.5-VL, and expand the caption to explicitly condition on these attributes while fine-tuning the generative model. Later, they can randomly select from the sets to marginalize out their effects.

They present main results on 5 datasets, two few-shot settings, three backbones, and attached to seven existing data generation techniques, showing gains in most settings. They also use two long-tailed datasets. For additional analysis, they compare distributions, explore several captioning models, and include an ablation on the marginalization.

**Strengths:**

S1) The results show consistent improvement.

S2) A wide range of settings is considered, which provides good coverage of base model and several numbers of shots.

S3) The methodology is straightforward and well-explained.

S4) The paper is written clearly.

**Weaknesses:**

W1) (related to Q1) There are specific downsides to the chosen evaluation setting, specifically with the datasets chosen when compared to what is done in other work (e.g. Kim 2024, also [A]). The five datasets in the chosen setting are only fine-grained datasets--this is much less comprehensive than the 10 datasets used in the other setting. In the other setting, there are also more general datasets (e.g. ImageNet, Caltech101), some out-of-distribution datasets (EuroSAT, DTD).

W2) The work seems to under-emphasize the effect of the underlying generative model. Specifically, (a) the related work section misses discussion of the underlying text-to-image models and (b) the chosen generation models (SDv1.5 and SDv2.1) are far from state-of-the-art, given that many much stronger models are now available (e.g. SDXL, SDv3, FLUX, QwenImage). Given that synthetic data effectiveness is highly impacted by the underlying generative model, this would would be much more relevant to the current state if it explored or discussed more modern generative models.

W3) (not a major flaw, but could be better) The dataset chosen in the long-tail analysis section seems misaligned with the purpose, which appears to be showing scaling. While scaling the number of images, this dataset also changes classes, which is much less easy to compare. It would be cleaner to use the same classes and scale the number of images. The dataset choice can also be a bit misleading for readers, who might expect a long-tail dataset to be providing insights into where the method is better / worse (which are missing, as it is rather uniform).

W4) The takeaway in the section "is it distillation" seems a bit stronger than the provided results. It is conceivable that the model is distilling knowledge, but there are other factors bottlenecking the process further. The whether this is distillation is also a theoretical framework--if it is not distillation, it would be stronger to provide an alternative framework.

Small notes
w1) In Table 2, Flower-LT, Many, the choice to pick one 100 for bolding and the other for underlining is potentially misleading. It would be more clear to use the same notation.


[A] Diversified in-domain synthesis with efficient fine-tuning for few-shot classification, da Costa et al.

**Questions:**

Q1) Why did you choose the setting from Wang & Chen 2025, as opposed to other settings like Kim et al. 2024? (related to W1)

---

### Official Review · Reviewer_Rnaj · 2025-10-26

**Soundness:** 2
**Presentation:** 3
**Contribution:** 2
**Rating:** 4
**Confidence:** 3

**Summary:**

The paper proposes BOB (Beyond Objects), a two-stage framework for few-shot text-to-image (T2I) data generation. In the first stage, context preservation, the T2I model is fine-tuned with enriched captions containing class names, backgrounds, and poses extracted by a vision-language model. In the second stage, context marginalization, backgrounds and poses are randomly recombined across the dataset to break class-context correlations. Experiments on multiple fine-grained and long-tail datasets show consistent improvements over seven baselines, demonstrating the method’s effectiveness in generating diverse and less biased synthetic data.

**Strengths:**

1.The paper addresses an important and practical problem, reducing spurious correlations in few-shot text-to-image (T2I) data generation, and demonstrates clear performance gains on fine-grained and long-tail classification tasks.

2.The proposed two-stage framework (context preservation + context marginalization) is conceptually simple, easy to implement, and yields consistent improvements across datasets and backbones.

**Weaknesses:**

1.The method feels largely like an enhanced prompt optimization pipeline. Although the authors combine LoRA fine-tuning (context preservation) with dataset-level randomization (context marginalization), the overall novelty is limited.

2.The causal explanation between foreground and background is not new — similar causal interpretations (e.g., back-door adjustment) have appeared in previous few-shot or domain generalization literature.

3.It would be more convincing if the proposed method were demonstrated as a plug-and-play component applicable to existing baselines (e.g., Diff-II, DataDream), instead of only testing on SD v1.5/v2.1.

**Questions:**

1.In Figure 1, the extracted “pose” appears to be semantic information about the scene (e.g., flying, landing, on water) rather than geometric pose?

2.Could you provide an ablation on the number of synthetic images per class?

---

### Official Review · Reviewer_4gDk · 2025-10-28

**Soundness:** 2
**Presentation:** 2
**Contribution:** 2
**Rating:** 2
**Confidence:** 4

**Summary:**

The paper proposes BOB (Beyond OBjects), a fine-tuning and generation strategy to create synthetic training data for fine-grained classification from T2I models. The key ideas are: (1) Context preservation during T2I fine-tuning, which extracts class-agnostic attributes (background, pose) with a captioning model and injects them into per-image text prompts so the model learns to faithfully render context without overfitting to few exemplars; and (2) Context marginalization at generation: when synthesizing images, randomly sample background/pose pairs from a global caption bank (across classes) approximating an intervention that breaks spurious class–context links. Across Aircraft, CUB, Cars, Pets and three backbones (CLIP-B/16, ResNet-50, MAE-B/16), BOB improves few-shot accuracy and also helps in long-tail regimes.

**Strengths:**

1.  It presents a clear causal framing for removing spurious context that integrates with existing SD backbones.

2. The experiment is conducted on multiple datasets and backbones to demonstrate its good performance.

**Weaknesses:**

1. BOB relies on captions extracted from only 5–10 real exemplars per class using a VLM (Qwen-VL-7B) to describe background and pose attributes. However, since modern large language models already possess rich world knowledge about object appearances and environments, this dependence on limited real images could be avoided. By prompting an LLM directly (e.g., “Describe possible backgrounds and poses for an aircraft image”), one could construct a large and diverse attribute bank without relying on the few available examples. Such a strategy would naturally produce richer contextual variations (e.g., aircraft on a lawn, snowfield, or desert) that text-to-image models can easily render. Even if some generated contexts are physically implausible, the resulting diversity could significantly enhance classifier robustness and help it focus on the object foreground rather than dataset-specific correlations.

2. Although the paper presents a causal formulation based on the back-door adjustment via context marginalization, there is no empirical verification of the claimed causal effect. The authors assert that random sampling of background and pose achieves P(X∣do(Y)), but they do not quantify whether spurious correlations between class labels and contextual cues are actually reduced. Analyses such as mutual-information estimation, causal discovery, or counterfactual tests could provide such evidence. In addition, Modeling Z as just background+pose may be an oversimplification. Other nuisance variables (e.g., surrounding objects or complex scene elements) can still carry class-specific signals. Without sufficiency or sensitivity analyses on the definition of Z, the causal justification remains largely conceptual rather than empirically grounded.

3.  Overall, despite solid experimental execution and clear motivation, the novelty and causal rigor are limited. The core idea, i.e., sampling cross-class contexts, resembles a heuristic augmentation strategy rather than a verified causal intervention.

**Questions:**

1. Could the authors explain why the background and pose attributes must be extracted from the limited 5–10 real exemplars rather than generated directly via prompting a large language model? Would using an LLM to describe possible backgrounds and poses for each class (e.g., “Describe possible backgrounds and poses for an aircraft”) yield richer contextual diversity and potentially better generalization?

2. Have the authors considered constructing a larger background/pose bank, including imaginative or less realistic scenes (e.g., aircraft on snow or desert), to improve synthetic diversity? Would such diversity benefit classifier robustness even if some contexts are implausible?

3. How do the authors empirically verify that random sampling of background and pose actually implements the intended intervention P(X∣do(Y))? Are there any quantitative analyses (e.g., reduction in mutual information between class and background, causal discovery, or counterfactual tests) to support the causal claim?

4. Why is Z defined solely as {background, pose}? Have the authors examined whether additional nuisance factors (e.g., nearby objects or composition) also correlate with class identity and thus need to be marginalized?

---

### Official Review · Reviewer_jwHB · 2025-10-29

**Soundness:** 3
**Presentation:** 3
**Contribution:** 1
**Rating:** 2
**Confidence:** 4

**Summary:**

To generate data for fine-grained classification, the paper proposes a method called BOB to guide text-to-image (T2I) generation. BOB first extracts class-agnostic attributes from real images and fine-tunes the T2I model. It then generates images with random attributes to augment data for fine-grained classification. Experiments demonstrate that this method is effective.

**Strengths:**

- The paper is well-motivated.
- The paper is readable.
- Numerous experiments demonstrate that this method is effective.

**Weaknesses:**

1. The BOB method is too simple and lacks novelty. Obtaining more detailed captions and combining image elements to generate new images is rather trivial.

2. If the T2I model directly generates synthetic data without fine-tuning—by simply adjusting prompts or random seeds—to augment real data, how would the downstream classification performance compare?

3. In Table 1, are the training epochs for the “Read only” method and the “BOB” model the same? If so, it means that “Read only” was trained for fewer steps, which is unfair. A fair comparison should ensure both methods are trained for the same number of steps.

**Questions:**

see weakness.

---

### Note · Authors · 2025-11-13

I have read and agree with the venue's withdrawal policy on behalf of myself and my co-authors.